# Energy-Based Discrete Mask Approximation for 3D Molecular Graph Explanation

## Abstract

In recent years, Graph Neural Networks (GNNs) have become a powerful tool for modeling molecular data. To enhance their reliability and interpretability, various explanation methods have been developed to identify key molecular substructures, specifically a set of edges, in the decision-making process. Early work with 2D GNNs represented molecules as graphs with atoms as nodes and bonds as edges, neglecting 3D geometric configurations. While existing explanation methods perform well on 2D GNNs, there is a pressing need for 3D explanation methods tailored for 3D GNNs, which outperform 2D GNNs in many tasks. Current explanation methods struggle with 3D GNNs due to the construction of edges based on cut-off distances in 3D GNNs, resulting in an exponentially large number of edges. We identify the sources of errors in explanations and decompose them into two components based on a derived upper bound between the optimized masks and the actual explanatory subgraph. This gap can be significant, especially for 3D GNNs because of the large number of edges. To achieve optimal explanation fidelity, our method aims to bridge this gap by assigning two energy values to each atom based on its contribution to the prediction: one energy reflects the scenario where this node is important in making the decision, while the other represents the scenario where it is unimportant. In analogy to physics, lower energy values indicate greater stability in the prediction, and thus, we are more confident about the scenario with which it is associated. Our approach strives to push up and down the energies, respectively, to distinguish these two scenarios to simultaneously minimize both components of the derived upper bound of error, enabling us to identify a stable subgraph that maintains high explanation fidelity. Experiments conducted on backbone networks and the QM9 dataset demonstrate the effectiveness of our method in providing accurate and reliable explanations for 3D graphs.

## 1 Introduction

In recent years, molecular learning has emerged as a crucial area of study, driving advances in drug discovery, protein engineering, and materials science (Gori et al., 2005; Wu et al., 2018; Shervashidze et al., 2011; Fout et al., 2017). Traditionally, molecules have been represented as 2D planar graphs, where atoms serve as nodes and chemical bonds are depicted as edges without considering the geometric configurations. The limitations of 2D representations in capturing molecular properties have led to a growing focus on 3D graph representations (Kipf & Welling, 2017; Defferrard et al., 2016; Veličković et al., 2018; Zhang et al., 2018; Xu et al., 2019; Gao et al., 2021) that represent entities with spatial coordinates, enabling them to capture complex spatial dependencies that are critical for tasks involving 3D molecular structures. This shift is critical because the 3D structure of molecules, particularly their spatial arrangement, directly influences their chemical behavior and biological functions. In response, 3D GNNs have been developed to incorporate geometric information and shown to outperform their 2D counterparts in numerous tasks (Schütt et al., 2017; Satorras et al., 2021; Gasteiger et al., 2020b; Liu et al., 2022; Shuaibi et al., 2021; Thomas et al., 2018; Liao & Smidt, 2022; Anderson et al., 2019; Fuchs et al., 2020; Schütt et al., 2021; Batzner et al., 2022).

As GNNs have shown great results in molecular learning, the need for explainability and interpretability has become increasingly important. Molecular systems are inherently complex, and GNNs are often treated as black-box models, making it difficult to understand how specific structural features contribute to predictions, which raises significant concerns regarding transparency in

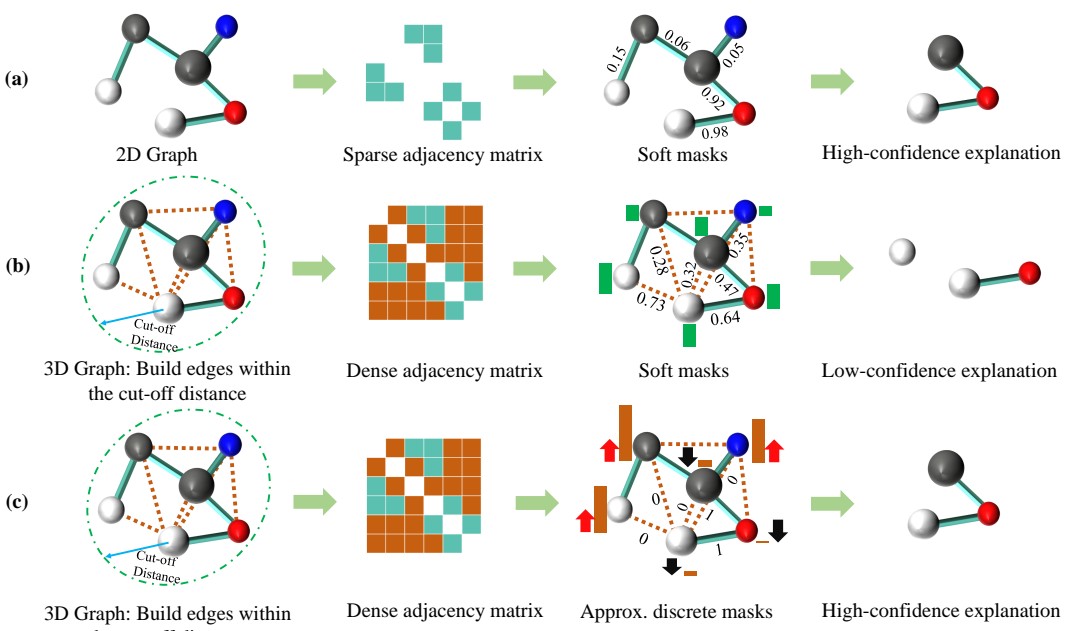

Figure 1: An illustration of the structural differences between 2D and 3D GNN explanations, as well as the challenges posed by existing methods. In the third column, the bars represent the soft masks for nodes, while the numbers correspond to the edge masks derived from the energies of the nodes. Specifically, due to the differing assumptions in 3D GNNs, our aim is to identify a subset of nodes for the explanation. However, using soft masks typically results in explanations of low confidence, which undermines explanation fidelity. To address this issue, we assign energies to each node. By pushing up and down these energies, we can obtain approximately discrete masks that provide more confident explanations. Further details are provided in Sec. 3.

the decision-making process. GNN explanation methods aim to illuminate these decision-making processes by identifying key substructures of the graph (the molecule) that influence the model's predictions. The primary objective is to extract a compact subgraph composed of a limited number of edges or nodes that effectively represent the behavior of the original graph. Groundbreaking research has addressed these challenges, advancing our understanding of graph learning mechanisms across various contexts (Ying et al., 2019; Yuan et al., 2020; Shrikumar et al., 2017; Luo et al., 2020; Yuan et al., 2021; Pope et al., 2019b; Schwarzenberg et al., 2019; Huang et al., 2023).

While existing explanation methods have proven effective for 2D GNNs, there is an urgent need for explanation techniques specifically designed for 3D GNNs. Current explanation methods face challenges with 3D GNNs due to the construction of edges based on cut-off distances, leading to an exponentially large number of edges (Schütt et al., 2017; Gasteiger et al., 2020b;a; Wang et al., 2022; Liu et al., 2022; Satorras et al., 2021). In our study, we identify the sources of errors in explanations and break them down into two components, informed by a derived upper bound that relates the optimized masks to the actual subgraph. This gap is particularly significant for 3D GNNs due to the large volume of edges involved. In 2D GNN explanations, there are typically at most a few edges with mask values around $0.5$, indicating uncertainty about their inclusion or exclusion in the final explanatory graph. However, in 3D GNN explanations, the number of certain edges grows rapidly, leading to suboptimal results and ambiguity in explanation results that complicate the decision-making process rather than explaining it.

To enhance explanation fidelity, our method aims to bridge this gap by assigning two energy values to each atom in the molecular graph. One energy reflects the scenario where this node is important in making the decision, while the other represents the scenario where it is unimportant. Drawing an analogy to physics (Rupp et al., 2012; Schütt et al., 2017), we assert that nodes with lower energy values correspond to greater stability in the explanatory results; thereby, we are more confident about the scenario with which it is associated. Current explanation models (Ying et al., 2019; Luo

et al., 2020; Miao et al., 2022b) only optimize the first term in our derived bound leading to over-smoothing of the soft masks. Our approach seeks to push the lower energy down and push the higher energy up to simultaneously optimize both components of the derived error bound, thereby reducing discrepancies between the identified explanatory subgraph and the associated edge masks. By achieving a lower energy state, we can accurately and confidently identify a stable subgraph that exhibits high explanation fidelity. Our method addresses the unique challenges posed by 3D data structures and the complex relationships among atoms, which contribute to the exponential growth of edge connections. An illustration of our method to mitigate challenges caused by the key structural differences between 2D and 3D GNN explanations is presented in Fig. 1. Experiments conducted on several backbone networks and the QM9 dataset (Ramakrishnan et al., 2014) validate the efficacy of our method, demonstrating its capacity to deliver accurate, stable, and reliable explanations for 3D graphs.

We summarize our contributions as follows:

- Leveraging the structural differences between 2D and 3D GNNs, we reformulate graph explanations specifically for 3D GNNs.

- We establish an error bound for graph explanations, dividing them into two components: the first is the focus of existing methods, while the second has long been overlooked.

- Based on the derived upper bound, we introduce Energy-based Discrete Mask Approximation to address this bottleneck, optimizing both components simultaneously.

- Experimental results demonstrate that our method is effective and highly generalizable for explaining 3D GNNs.

## 2 BACKGROUND AND RELATED WORK

In this section, we begin by presenting the formal definition of the graph explanation task in Sec. 2.1, which establishes a conceptual framework for understanding various graph explanation methods. Following that, in Sec. 2.2, we provide a comprehensive review of the key methodologies that have been proposed to generate explanations in this context. Finally, Sec. 2.3 delves into the definition and formulation of Energy-Based Models (LeCun et al., 2006), outlining their role in enhancing the interpretability of GNNs and their applications in providing insights into molecular structures and behaviors.

### 2.1 GRAPH EXPLANATION

A 2D molecular graph $G$ is represented as $G = (\mathcal{V}, \boldsymbol{X}, E)$, where $\mathcal{V} = \{v_1, v_2, \ldots, v_n\}$ denotes a set of $n$ nodes, and $\boldsymbol{X} = [\mathbf{x_1}, \mathbf{x_2}, \ldots, \mathbf{x_n}]^T \in \mathbb{R}^{n \times d_v}$ is the node feature matrix, with each $\mathbf{x_i} \in \mathbb{R}^{d_v}$, where $d_v$ represents the dimension of the node features. Graph neural networks (GNNs) utilize the edge set $E = \{e_{ij} \mid i, j \in \mathcal{V} \text{ and } i \neq j\}$ to facilitate message passing and aggregation between nodes. The edge $e_{ij} \in \{0, 1\}$ denotes whether there is an edge from node $i$ to node $j$, and the adjacency matrix $\mathbf{A} \in \{0, 1\}^{n \times n}$ is used to indicate the presence or absence of edges between all pairs of nodes. A graph model $\Phi$ is a mapping from a graph $G$ to a prediction $\hat{Y}$ in relation to the target variable $Y$. This target can represent discrete labels in a graph classification task or continuous values in a regression task. In this study, we specifically concentrate on graph regression tasks without loss of generality.

**Graph Explanation:** Following the definition in Ying et al. (2019), the objective of instance-level graph explanation is to identify a subgraph $G_S \subseteq G$ that is important to the target $Y$. This is formally expressed as:

$$G_S^* = \arg\min_{G_S \subseteq G} \mathcal{L}(Y; \Phi(G_S)) \quad \text{s.t.} \quad |G_S| \leq B, \tag{1}$$

where $\mathcal{L}$ denotes the task-dependent loss function, and $B$ represents a size constraint on the subgraph to avoid trivial solutions. Eq. (1) can be rewritten as:

$$G_S^* = \arg\min_{\mathbf{M}} \mathcal{L}(Y; \Phi(\mathbf{X}, \mathbf{M} \odot \mathbf{A})) \quad \text{s.t.} \quad \mathbf{M} \in \{0,1\}^{n \times n}, \quad \sum_{i=1}^{n}\sum_{j=1}^{n}\mathbf{M}_{ij} \le B. \quad (2)$$

Directly solving Eq. (2) leads to a computationally intractable combinatorial optimization problem with complexity $O(2^n)$. Exiting works relaxed the discrete (hard) masks with discrete values 0 and 1 to soft masks with values between 0 and 1:

$$G_S^* = \arg\min_{\mathbf{M}'} \mathcal{L}(Y; \Phi(\mathbf{X}, \mathbf{M}' \odot \mathbf{A})) \quad \text{s.t.} \quad \mathbf{M}' \in [0,1]^{n \times n}, \quad \sum_{i=1}^{n}\sum_{j=1}^{n}\mathbf{M}'_{ij} \le B. \quad (3)$$

Such relaxation enables gradients to be back-propagated; thus, gradient descent can be used to efficiently solve this problem.

## 2.2 EXISTING EXPLANATION METHODS

Graph Neural Network (GNN) explanation methods can be classified according to several criteria: transductive or inductive explanations, instance-level explanations (Ying et al., 2019; Schlichtkrull et al., 2021; Wang et al., 2021) versus model-level explanations (Yuan et al., 2020), model-specific approaches (Dai & Wang, 2021; Miao et al., 2022a; Pope et al., 2019a) compared to model-agnostic methods (Luo et al., 2020; Yuan et al., 2021; Zhang et al., 2021), and node-level (Ying et al., 2019; Pope et al., 2019a) versus graph-level explanations (Wang et al., 2021; Yuan et al., 2020). In terms of explanation strategies, four primary categories emerge: (1) Gradient-based methods (Shrikumar et al., 2017; Zhou et al., 2016; Baldassarre & Azizpour, 2019) compute the gradients of target predictions with respect to inputs via back-propagation but often impose structural constraints on GNNs; (2) Decomposition methods (Pope et al., 2019b; Schwarzenberg et al., 2019; Schnake et al., 2021) assign importance scores to input features by analyzing model parameters to reveal relationships between inputs and outputs; (3) Surrogate methods (Huang et al., 2023; Zhang et al., 2021) use interpretable models to explain the behavior of complex GNNs, though they often encounter difficulties with the discrete and topological nature of 3D graphs; (4) Perturbation-based methods (Ying et al., 2019; Luo et al., 2020; Yuan et al., 2021) identify important subgraphs by perturbing edges or nodes with masks and analyzing output prediction changes.

## 2.3 ENERGY-BASED MODEL

The central concept of Energy-Based Models (EBMs) is the use of Boltzmann distributions to assess the likelihood of input samples. This involves defining a function $\mathcal{E}(\mathbf{x_i}) : \mathbb{R}^{d_v} \to \mathbb{R}$ that assigns a non-probabilistic scalar known as *energy* to each configuration of the input data. Influential works (Xie et al., 2016; 2018) have significantly shaped research in this domain. EBMs have achieved notable success in various applications, including classification (Li et al., 2022; Grathwohl et al., 2019), regression tasks (Danelljan et al., 2020), structured prediction (Belanger & McCallum, 2016; Rooshenas et al., 2019), and out-of-distribution (OOD) detection (Liu et al., 2020; Wu et al., 2023). Additionally, (Yu et al., 2022; Pang & Wu, 2021) have explored the use of EBMs in latent space for generation, while others have applied them to unsupervised learning (Ranzato et al., 2007) and concept-based modeling (Xu et al., 2024).

## 3 ENERGY-BASED DISCRETE MASK APPROXIMATION

In this section, we present the Energy-based Discrete Mask Approximation (EDMA), a principled approach designed for 3D GNN explanation. We begin by analyzing the differences between 2D and 3D graph explanations in Sec. 3.1. With such differences, we reformulate 3D graph explanations and identify an upper bound on the explanation loss in Sec. 3.2. This bound consists of two parts; while existing methods succeed in optimizing the first part, the second part is often overlooked. In Sec. 3.3, we provide a detailed presentation of our method to simultaneously optimize all the terms in the derived upper bound.

### 3.1 2D V.S. 3D GRAPH EXPLANATION

The primary distinction between 2D and 3D graph explanations arises from the structural differences inherent in 2D and 3D graphs. In 2D graphs, each node is associated with a set of edges that connect these nodes in a planar layout. The relationships and interactions among nodes are captured in this planar representation. In 3D graphs, nodes are represented in three-dimensional space, allowing for a more accurate depiction of the physical arrangement and spatial relationships between entities, while edges are typically determined from the coordinates of the nodes by a cut-off distance (Schütt et al., 2017; Gasteiger et al., 2020b;a; Wang et al., 2022; Liu et al., 2022; Schütt et al., 2021; Thomas et al., 2018).

More concretely, for small molecular structures, the number of bonds (edges) between atoms (nodes) is typically limited, resulting in a rather sparse graph. However, 3D GNNs do not utilize chemical bonds as edges; instead, the 3D spatial configurations of nodes are used to construct edges resulting in an exponentially large number of edges. As a result, 3D GNN explanation poses significant challenges to existing explanation methods, partially illustrated in Fig. 1 and detailed below:

1. **Differing Assumptions in 2D and 3D Explanations**: The assumptions underlying graph explanations differ between 2D and 3D graphs. For instance, in 2D graph explanations, methods such as those proposed by (Ying et al., 2019; Luo et al., 2020) assume that the graph being explained is a random graph (Gilbert, 1959; ERDdS & R&wi, 1959), with edges considered independent of one another. However, this assumption does not hold for complex networks like molecular dynamics systems, where force field theory models both intramolecular interactions and inter-molecular terms, contributing to the total energy $E$ of the molecule (Leach, 2001).

2. **Dense Adjacency Matrix**: The adjacency matrix $\mathbf{A}$ in 3D graphs is typically dense, unlike the sparse adjacency matrix commonly observed in 2D graphs. As one can imagine, this leads to a problem of combinatorial complexity with respect to the number of edges with discrete masks. Even with soft mask relaxation, the large number of edges introduces a substantial lack of confidence in identifying the explanatory subgraph, often resulting in suboptimal explanation outcomes. Specifically, with low confidence in distinguishing important and unimportant sub-parts, the soft-masked "subgraph" deviates a lot from the final discrete explanatory sub-graph. The optimization process might find a soft-masked "sub-graph" with minimal loss in Eq. (3); however, when we decide the explanatory subgraph from soft-masks, the lack of confidence in the soft masks leads to poor final explanation performance.

### 3.2 REFORMULATING 3D GRAPH EXPLANATION

The inputs to 3D GNNs consist of nodes with 3D spatial coordinates, and edges are constructed based on this spatial information. The common assumption of random graph structures no longer holds in this context. Simply applying current graph explanation methods without accounting for the structural differences is unlikely to yield explanations of scientific meanings.

*As discussed in the first challenge outlined in Sec. 3.1, we should define the explanatory substructure to be a subset of nodes.* To this end, we place masks on the nodes, which are then transformed into edge masks. For node $i$, there will be an associated soft-mask value

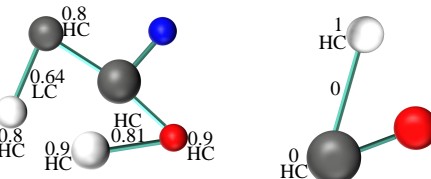

**Soft mask:** Low Confidence  **Discrete mask:** High Confidence

Figure 2: A comparison between soft masks and discrete masks, denoting **HC** for High Confidence and **LC** for Low Confidence. The edge masks used for message passing are constructed from node masks. However, soft masks can lead to great discrepancies between the optimization objective and the final explanatory substructure, as indicated in Eq. (5).

$m_i \in [0, 1]$, and $\mathbf{m} \in [0, 1]^n$ denotes the set of all node masks. The edge masks are then constructed by $\mathbf{M}' = \mathbf{m} \otimes \mathbf{m}$, where $\otimes$ denotes the outer product. Specifically, Eq. (3) can be rewritten as

$$G_S^* = \underset{\mathbf{M}'}{\arg\min} \, \mathcal{L}(Y; \Phi(\mathbf{X}, \mathbf{M}' \odot \mathbf{A})) \text{ s.t. } \mathbf{m} \in [0, 1]^n, \quad \mathbf{M}' = \mathbf{m} \otimes \mathbf{m}, \quad \sum_{j=1}^{n} \mathbf{m}_i \leq K, \quad (4)$$

where $K$ is the budget on the number of nodes in the final explanatory substructure. To this end, we would like to introduce a shortcoming in relaxing discrete masks to soft-masks: The discrepancy between the optimized soft-masked "subgraph" and the final explanatory subgraph. Mathematically,

$$G_S^* = \arg\min_{G_S \subseteq G} \mathcal{L}(Y; \Phi(G_S)) \leq \underbrace{\mathcal{L}(Y; \Phi(\mathbf{X}, \mathbf{M}' \odot \mathbf{A}))}_{\text{soft-mask explanation loss}}$$
$$+ \underbrace{\mathcal{L}(\Phi(\mathbf{X}, \mathbf{M}' \odot \mathbf{A}); \Phi(\mathbf{X}, \mathbf{M} \odot \mathbf{A}))}_{\text{discrepancy between soft and discrete masks}}. \quad (5)$$

In this bound, the first term represents the soft-mask explanation loss in the relaxed optimization Eq. (3), which we solve through gradient descent. The second term depicts the discrepancy between soft and discrete masks, and this has been overlooked in existing GNN explanation methods. Existing studies term this issue as "introduced evidence". Any value in masks that is not strictly zero or one can introduce new semantics or noise into the explanation, potentially impacting the results (Dabkowski & Gal, 2017; Lin et al., 2021). For instance, even if the value of $\mathbf{M}'_{ij}$ is small, the edge $e_{ij}$ may still facilitate message passing between node $i$ and $j$. We will refer to this as confidence of the soft masks, where a mask value close to 0 or 1 indicates high confidence about the substructure's contribution to decision-making process. *While this bound generally applies to both 2D and 3D GNNs, 3D GNNs suffer much more from this issue for reasons given in the second challenge outlined in Sec. 3.1.* In 3D GNNs, there are exponentially many edges, and the accumulation of information passed during message passing can significantly influence the explanation results even with edge masks of high confidence. Even worse, due to the intrinsic nature of 3D GNNs, the node masks will largely decrease the confidence and stability in final explanation as illustrated in Fig. 2. To this end, we are ready to present the Energy-based Discrete Mask Approximation method to mitigate this issue.

### 3.3 EDMA FOR CONFIDENT 3D GRAPH EXPLANATION

We now present our method EDMA for confident 3D graph explanation that simultaneously minimizes both terms in Eq. (5).

Instead of using soft masks for the selection of explanatory nodes, we treat the selection of nodes as states within a system, where the energy levels of these states determine their probability of being part of the explanatory subgraph. The EBM function $\mathcal{E}(\mathbf{e_i}) : \mathbb{R}^d \to \mathbb{R}$ maps the node embedding $\mathbf{e_i}$ to a scalar value known as *energy*. Following Liu et al. (2020), the energy for a node with respect to class $c$ is defined as $\mathcal{E}(\mathbf{e_i}, c) = \mathcal{E}_c(\mathbf{e_i}) = \frac{-\phi_c(\mathbf{e_i})}{T}$, where $\phi_c$ extracts logits for class $c$ and $0 < T < 1$ serves as a control hyper-parameter analogous to the temperature in the physics. It will push up the larger energy and push down the smaller energy. To see this, suppose $\phi_0 = 2$ and $\phi_0 = 5$ then the difference between them is 3. With $T = 0.1$, the energies will be 20 and 50, respectively, and the difference between them is 30 now. With a smaller value of $T$, we further

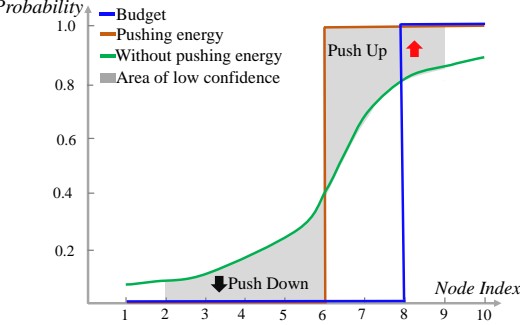

Figure 3: An illustration of the effects of the explainer function $f$. The node indices are arranged based on their probability values. By pushing up and down the energies, the masks become approximately discrete, enhancing confidence in the explanatory substructure. Moreover, varying the values of the stretching parameters ($\gamma, \zeta$ in Eq. (6)) enables us to better control the budget.

amplify the difference between these two energies, leading to more confident explanation with our explainer function as defined in Eq. (6). This process is analogous to the temperature in physics, when $T$ is small, the system is in a low-energy state, leading to probabilities closer to 0 or 1; in other words, a more confident selection of nodes.

Then, an explainer function $f$ is used to compute the probabilities that each node $i$ belongs to the explanatory subgraph, represented as $P_i(c = 1) = f(\mathcal{E}_0(\mathbf{e_i}), \mathcal{E}_1(\mathbf{e_i}))$. Inspired by the hard concrete distribution (Louizos et al., 2017), $f : \mathbb{E} \times \mathbb{E} \mapsto [0, 1]$, with $\mathbb{E}$ being the potential energy state space

for all nodes, is a function that takes both energies and produces a single scalar value indicating probabilities defined as

$$f\big(\mathcal{E}_\mathbf{0}\,(\mathbf{e_i}), \mathcal{E}_1\,(\mathbf{e_i})\,\big) = \min\left(1, \max\left(0, \frac{1}{1 + e^{(\mathcal{E}_\mathbf{0}(\mathbf{e_i}) - \mathcal{E}_1(\mathbf{e_i}))}}\,(\zeta - \gamma) + \gamma\right)\right), \qquad (6)$$

where $\gamma < 0$, $\zeta > 1$ are hyper-parameters to stretch the probability to the interval $(\gamma, \zeta)$ and then truncate the value to the range $[0, 1]$. Together with the temperature, this will help us obtain more confident explanations in terms of having all the probabilities close to either 0 or 1 and better control the budget $K$ as we can set appropriate values of $\gamma$ and $\zeta$ to obtain the desired number of probabilities closer to 1. In Fig. 3, we illustrate how our energy-based explainer function generates approximately discrete masks while effectively managing the budget. Our explainer function takes energy values as input and outputs the probability that a node belongs to the explanatory substructure. By increasing the energies of important nodes and decreasing those of unimportant ones, we enhance their distinction, resulting in approximately discrete masks that yield more confident explanatory substructures. Furthermore, the stretching parameters in Eq. (6) regulate the number of nodes for which energies are pushed up.

Without loss of generality, we denote $m_i = f\big(\mathcal{E}_\mathbf{0}\,(\mathbf{e_i}), \mathcal{E}_1\,(\mathbf{e_i})\,\big)$ as the explanation mask. Finally, the mutual information term in Eq. (4) and the explainer function in Eq. (6) are then jointly optimized to classify nodes and determine whether they belong to the explanatory subgraph. The final optimization function for our proposed method is as follows:

$$\mathcal{L}_{final} = \mathcal{L}(Y; \Phi(\mathbf{X}, \mathbf{M}^{'} \odot \mathbf{A}, \mathbf{r})) + \alpha \| f(\mathcal{E}_\mathbf{0}(\mathbf{e_i}), \mathcal{E}_1(\mathbf{e_i})))\|_1, \qquad (7)$$

where $\alpha$ is a parameter that balances the information loss and the explainer function loss. With this particular formulation, we simultaneously optimize both terms in our derived bound in Eq. (5), leading to approximately discrete, i.e., confident, probabilities for the inclusion or exclusion of a certain node in the final explanatory subgraph.

## 4 EXPERIMENTAL STUDIES

We begin by outlining the experimental setup in Sec. 4.1, where we provide details on the dataset, baseline methods, and evaluation metrics. Sec. 4.2 presents a comparative analysis of the quantitative results of our method against baseline approaches. In Sec. 4.3, we offer a qualitative analysis to further illustrate the interpretability and effectiveness of the proposed method. Finally, an ablation study is conducted in Sec. 4.4 to evaluate the contributions and significance of various components in our approach.

### 4.1 EXPERIMENTAL SETUP

**Dataset.** In this work, we utilize the widely adopted QM9 dataset (Ramakrishnan et al., 2014), a comprehensive 3D molecular dataset frequently used to predict various molecular properties. We specifically use the QM9 version available in PyTorch Geometric (PyG), along with its predefined training and test splits. As backbone models for $\Phi$, we adopt the pretrained SchNet (Schütt et al., 2017) and DimeNet++ (Gasteiger et al., 2020b;a) architectures, both well-suited for 3D graph-based tasks. Our study targets the prediction of two key properties: the dipole moment ($\mu$) and the free energy at 298.15K (denoted as $G_f$ to avoid confusion with the graph notation $G$).

**Baselines.** We compare our approach against several state-of-the-art baselines. GNNExplainer (Ying et al., 2019) and PGExplainer (Luo et al., 2020) are leading explanation methods for 2D GNNs, designed for transductive and inductive tasks, respectively. However, due to structural differences discussed in Sec. 3.1, these methods are not directly applicable to 3D GNNs. To adapt them for 3D molecular graphs, we place masks on nodes, generate edges in a manner similar to our approach, and use these to perturb node embeddings and generate explanations. These adapted methods, referred to as GNNExplainer-Dense and PGExplainer-Dense, serve as key baselines for evaluating performance on the QM9 dataset.

In addition, we include LRI (Miao et al., 2022b) in our comparisons, as it is currently the only method specifically designed for geometric graph explanations. We employ the LRI-Bernoulli variant, which identifies key nodes relevant to downstream regression tasks, making it a strong baseline

Table 1: Explanation fidelity for both baseline methods and our propose EDMA method regarding the property $\mu$ (dipole moment) is presented using SchNet. The best results are highlighted in bold.

| Top-$k$ | 2 | 3 | 4 | 5 | 6 | 7 | 8 | 9 |
|---|---|---|---|---|---|---|---|---|
| GNNExplainer-Dense | 3.88 | 5.62 | 7.28 | 8.05 | 8.27 | 8.00 | 7.59 | 6.87 |
| PGExplainer-Dense | 2.91 | **3.73** | 4.83 | 6.09 | 6.62 | 6.55 | 6.81 | 6.08 |
| LRI-Bernoulli | 3.50 | 4.84 | 6.16 | 6.88 | 7.10 | 7.29 | 7.43 | 7.32 |
| **EDMA** | **2.74** | **3.73** | **4.31** | **4.83** | **5.08** | **5.47** | **5.72** | **5.31** |

Table 2: Explanation fidelity for both baseline methods and our proposed EDMA method regarding the property $G_f$ (free energy at 298.15K) is presented using SchNet. The best results are highlighted in bold.

| Top-$k$ | 2 | 3 | 4 | 5 | 6 | 7 | 8 | 9 |
|---|---|---|---|---|---|---|---|---|
| GNNExplainer-Dense | 9.66 | 8.48 | 7.24 | 6.03 | 4.78 | 3.51 | 2.26 | 1.09 |
| PGExplainer-Dense | 10.26 | 9.56 | 8.68 | 7.74 | 6.52 | 5.40 | 4.21 | 3.94 |
| LRI-Bernoulli | 9.39 | 8.39 | 7.38 | 6.59 | 5.93 | 5.31 | 4.78 | 4.09 |
| **EDMA** | **8.66** | **7.45** | **6.23** | **5.07** | **3.74** | **2.55** | **1.36** | **0.21** |

for explaining 3D molecular graph data. All baseline methods are implemented using PyG with necessary adjustments to ensure consistency in the experimental setup. Further details are provided in Appendix A.

**Evaluation Metrics** Following the standard protocol for QM9 data, we use Mean Absolute Error (MAE) to evaluate the performance of 3D GNN predictions against ground-truth molecular properties. A lower MAE indicates higher predictive accuracy. In the context of explaining 3D molecular graphs, let $MAE_W$ represent the prediction error using the entire graph, while $MAE_S$ denotes the prediction error using the optimal subgraph selected for explanation, as described in Eq. (2). Naturally, $MAE_S$ is expected to be higher than $MAE_W$, as the complete graph generally yields better predictions than the subgraph, given the same pretrained 3D GNN model (such as SchNet or DimeNet++). We define explanation fidelity as $Fidelity^- = MAE_S - MAE_W$, which measures the quality of explanations produced by different methods. *A lower Fidelity$^-$ indicates that the method provides a more accurate and reliable explanation.* It is important to note that the reported results represent the average across all molecules in the QM9 test set. Since the standard deviation is two orders of magnitude smaller than the average fidelity, we do not report the standard deviation in our results.

### 4.2 COMPARISON RESULTS

We demonstrate the effectiveness of our method by comparing it to baseline approaches on the QM9 dataset. The results obtained using SchNet are presented in Tables 1 and 2. Notably, all baseline methods select the top-$k$ nodes as explanations, with $k$ ranging from 2 to 9. Our method consistently outperforms these baselines, producing explanatory subgraphs with the lowest explanation fidelity (where lower fidelity indicates better performance). This suggests that by optimizing the two distinct components based on the derived upper bound, we achieve a closer alignment between the desired discrete masks and the approximate discrete masks generated via the energy-based model (EBM). By appropriately controlling the loss function, we can either increase or decrease the energy of each atom, amplifying the distances between explanatory and non-explanatory parts, making them easier to identify. Similar results are observed in Tables 3 and 4, which use DimeNet++ as the backbone.

### 4.3 QUALITATIVE RESULTS

In this section, we present qualitative results regarding explanation fidelity. Functional groups play a significant role in determining the chemical properties of molecules; thus, an explanation method that generates results with high fidelity is more likely to accurately identify these functional groups. We visualize the results using various methods on several real molecules from the QM9 dataset, focusing on the property $\mu$ with DimeNet++, as shown in Fig. 4. Additionally, we provide chemical explanations derived from domain knowledge and elaborate on the contributions of functional groups.

Table 3: Explanation fidelity for both baseline methods and our propose EDMA method regarding the property $\mu$ (dipole moment) is presented using DimeNet++. The best results are highlighted in bold.

| Top-$k$ | 2 | 3 | 4 | 5 | 6 | 7 | 8 | 9 |
|---|---|---|---|---|---|---|---|---|
| GNNExplainer-Dense | 2.50 | 2.29 | 2.07 | 1.80 | 1.53 | 1.27 | 1.04 | 0.82 |
| PGExplainer-Dense | 2.52 | 2.46 | 2.13 | 2.07 | 1.84 | 1.67 | 1.50 | 1.40 |
| LRI-Bernoulli | 2.58 | 2.42 | 2.22 | 2.04 | 1.87 | 1.69 | 1.51 | 1.40 |
| **EDMA** | **2.40** | **2.07** | **1.76** | **1.47** | **1.22** | **1.02** | **0.85** | **0.71** |

Table 4: Explanation fidelity for both baseline methods and our propose EDMA method regarding the property $G_f$ (atomization free energy) is presented using DimeNet++. The best results are highlighted in bold.

| Top-$k$ | 2 | 3 | 4 | 5 | 6 | 7 | 8 | 9 |
|---|---|---|---|---|---|---|---|---|
| GNNExplainer-Dense | 65.74 | 62.56 | 58.91 | 54.65 | 49.15 | 43.18 | 36.69 | 30.53 |
| PGExplainer-Dense | 64.32 | 61.25 | 55.33 | 51.64 | 46.13 | 40.33 | 34.35 | 29.41 |
| LRI-Bernoulli | 64.76 | 60.46 | 55.29 | 49.97 | 44.66 | 39.41 | 34.19 | 29.49 |
| **EDMA** | **63.83** | **59.42** | **54.48** | **49.26** | **43.84** | **38.15** | **33.12** | **28.47** |

For each molecule, we match the number of atoms to those in the chemical explanations and select the top-$k$ atoms across all methods for a fair comparison. Since functional groups are not necessarily connected, we do not impose a requirement for the explanatory subgraphs to be connected. Our results indicate that using EDMA to generate explanatory subgraphs enhances the likelihood of identifying the true explanatory components, which align more closely with established scientific knowledge and yield meaningful explanations. Overall, by adopting approximately discrete masks, our method provides more reasonable explanations that better reflect chemical understanding.

Figure 5: The distribution of masks generated by EDMA and EDMA-soft demonstrates that pushing energies to a greater extent results in approximately discrete masks. Further results shown in Table 5 indicate that approximately discrete masks yield superior performance.

## 4.4 ABLATION STUDY

We assert that the discreteness of masks significantly impacts the explanation results for 3D graphs. To validate this claim, we conducted an ablation study. To isolate the effects of the Energy-Based Model (EBM), we kept all other components the same while altering only the parameters related to energy adjustments. We refer to the variant where we push the energies to a greater extent as EDMA, and the one with less energy adjustments as EDMA-soft. First, we verified that our method generates approximately discrete masks. In Fig. 5, we illustrate the distributions produced by these two variants, revealing a clear distinction between them. Second, we demonstrate that EDMA enhances confidence in the explanation results, thereby boosting the performance and scientific significance of 3D explanations. We employed the same evaluation criteria as outlined in Sec. 4.1 and selected the top-$k$ nodes. The explanation results are presented in Table 5. The findings indicate that masks lacking confidence (i.e., not approxi-

Table 5: Experimental results comparing explanation fidelity of the EDMA method and its soft mask variant, EDMA-soft (achieved by adjusting hyper-parameters), are presented for the property $\mu$ (dipole moment) using the SchNet model.

| Top-$k$ | 2 | 3 | 4 | 5 | 6 | 7 | 8 | 9 |
|---|---|---|---|---|---|---|---|---|
| **EDMA** | 2.74 | 3.73 | 4.31 | 4.83 | 5.08 | 5.47 | 5.72 | 5.31 |
| **EDMA-soft** | 3.32 | 4.52 | 6.14 | 7.20 | 7.88 | 8.09 | 8.02 | 7.44 |

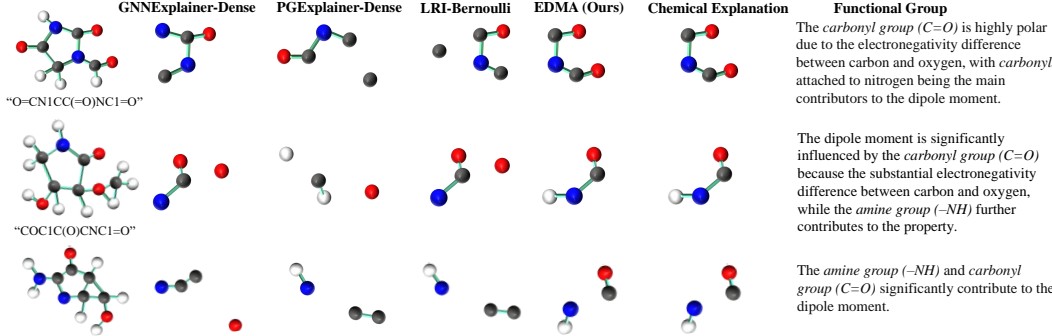

Figure 4: The first column showcases real molecules from the QM9 dataset, along with their corresponding SMILES strings. The following columns present the explanation results from various baseline methods alongside our EDMA method. Finally, the last two columns offer insights into the chemical explanations and the effects of functional groups associated with each molecule. **It is evident that our method, EDMA, delivers the most accurate explanation results, aligning closely with chemical priors.**

mately discrete) negatively impact explanation performance, whereas approximately discrete masks yield superior results as it reduces the second term in Eq. (5).

## 5 CONCLUSIONS AND FUTURE WORK

In conclusion, our research highlights the advancements in explaining 3D graphs. While existing explanation methods have made strides in interpreting 2D GNNs, there remains a critical gap in developing effective explanations for 3D GNNs due to the complexities introduced by the geometric configurations and the sheer volume of edges. By acknowledging the varying assumptions in 3D GNNs, we reformulate 3D GNN explanations and identify a bottleneck in all the existing methods for 3D explanations. We propose a novel energy-based explanation function to generate probabilities that are approximately discrete and highly confident. Our method effectively bridges the gap between optimized masks and actual explanatory subgraphs, leading to improved explanation fidelity. The results obtained from experiments on backbone networks and the QM9 dataset affirm the efficacy of our approach in providing accurate and reliable explanations for 3D graphs. Building on our derived bound that characterizes the discrepancies between the optimized masks and final explanations, it would be intriguing to explore whether more advanced methods can be developed from these new bounds. Such advancements could significantly enhance the accuracy and reliability of explanations in 3D GNNs, ultimately offering deeper insights into molecular data.

ETHICS STATEMENT

This research centers on the development and assessment of explanations for 3D GNNs in molecular learning, which serve as deep learning frameworks for modeling complex molecular systems. The study does not involve human subjects, personal data, or sensitive information that could pose privacy, security, or fairness concerns. Additionally, no potential conflicts of interest, legal compliance issues, or harmful applications have been identified in this work.

REPRODUCIBILITY STATEMENT

All baseline models used in this study were employed with minimal or no modifications from their original versions. The datasets utilized are all publicly accessible, and sufficient details have been provided to enable the reproduction of our work. Details on the hyper-parameter search and settings are provided in Appendix A. Upon acceptance of the paper, we will make all the source code and configuration files necessary to replicate our results available.

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

# A  DETAILED EXPERIMENTAL SETUP

To ensure a fair comparison between our proposed model and existing methods, we performed extensive hyperparameter tuning for both our approach and the baseline methods. For the baseline methods, we employed the grid search to systematically explore their respective hyperparameter spaces. Each model's performance was assessed based on the Mean Absolute Error (MAE) on the test set, with the objective of identifying the optimal parameter configurations.

**SchNet**. For property $\mu$: we tested GNNExplainer-Dense using the following parameter settings: the coefficient for size loss varied from $1.0$ to $5.5$ with a step size of $2.0$, the coefficient for entropy loss ranged from $0.1$ to $1.1$ with a step size of $0.4$, and the number of training epochs ranged from $50$ to $500$ with a step size of $200$. The optimal parameters were determined as follows: a size loss coefficient of $1.0$, an entropy loss coefficient of $0.9$, and $50$ training epochs. Similarly, for PGExplainer-Dense, the training parameters were set with a training epoch of $40$, a size loss coefficient of $30.0$, and an entropy loss coefficient of $1.6$. For the LRI-Bernoulli method, the training epoch was set to $50$, with a prediction loss coefficient of $5.0$ and an information loss coefficient of $1.0$. The EDMA model's training epoch was established at $300$, with the parameter $\alpha$ set to $1.0$.

For the property $G_f$, the following parameters were established for GNNExplainer-Dense: the coefficient for size loss was set to $300.0$, the coefficient of entropy loss was also set to $300.0$, and the number of training epochs was fixed at $300$. In the case of PGExplainer-Dense, we set the training epoch to $100$, with a coefficient of size loss of $520.0$ and entropy loss coefficient of $300.0$. For LRI-Bernoulli, the training epoch was established at $300$, with a prediction loss coefficient of $1.0$ and an information loss coefficient of $3.0$. The training epoch for EDMA was similarly set to $300$, with the parameter $\alpha$ assigned to a value of $500.0$. Additionally, due to the presence of a shortcut embedding preceding the final readout layer in the PyG implementation for SchNet on property $G_f$, the node mask was multiplied by this embedding layer to ensure the validity of the experimental setup.

**DimeNet++**. For the property $\mu$ on DimeNet++, we established the following for GNNExplainer-Dense: the coefficient for feature size loss was set to $1.5$, the coefficient of entropy loss was set to $0.5$, and the number of training epochs was fixed at $200$. In the case of PGExplainer-Dense, the training epoch was set to $150$, with an coefficient of size loss of $0.5$ and an coefficient of entropy loss of $2.5$. For LRI-Bernoulli, the training epoch was set to $500$, with a prediction loss coefficient of $1.0$ and an information loss coefficient of $0.5$. The training epoch for EDMA was similarly established at $300$, with the parameter $\alpha$ assigned a value of $3.0$.

For the property $G_f$, we established the following parameters for GNNExplainer-Dense: the coefficient of size loss was set to $0.5$, the coefficient of entropy loss was set to $5.0$, and the number of training epochs was fixed at $300$. For PGExplainer-Dense, we set the training epoch to $100$, with both the size and entropy loss coefficient set to $5.0$. In the case of LRI-Bernoulli, the training epoch was set to $500$, with a prediction loss coefficient to $1.0$ and an information loss coefficient of $5.0$. The training epoch for EDMA was similarly established at $500$, with the parameter $\alpha$ assigned a value of $8.0$. It is important to note that while we use the same notation for $G_f$, in the PyG package for DimeNet++ this property specifically refers to the atomization free energy.

