# OpenReview forum: "Energy-Based Discrete Mask Approximation for 3D Molecular Graph Explanation"
_ICLR.cc/2025/Conference — Submitted to ICLR 2025_

### Official Review · Reviewer_nDf6 · 2024-10-17

**Soundness:** 3
**Presentation:** 3
**Contribution:** 2
**Rating:** 5
**Confidence:** 4

**Summary:**

The paper addresses the problem of devising ad-hoc graph explainers for 3D molecular representation, as prior work focuses on 2D graphs. Along the way, authors propose to use two relevance scores, instead of just one as typically done. This allows them to provide more confident explanations, with explanation scores better aligning to a discrete mask, which is typically desirable.

**Strengths:**

- 3D graphs are likely to have unique features that need to be addressed with property techniques, also from an explainability perspective
- The presentation and the writing are good
- **Touches several overlooked aspects in many explainers**, such as the discreteness of the mask, and the difference between edge vs node masks
- The paper focuses on regression explanation, which **is often overlooked in the XAI literature**.

**Weaknesses:**

- The differences outlined in Section 3.1 **do not seem to be convincing enough to justify ad-hoc approaches for 3D graphs**. Specifically, *Point 1* hinges on the unsuitability of the edge independence assumption in 2D explanations. While I agree that edge independence is oftentimes a very limiting assumption, I acknowledge that it is unsuitable for many 2D tasks as well (consider for example the case of social interactions, 2D molecules, or transportation networks), therefore not constituting a major difference between 2D and 3D graphs. *Point 2*, instead, highlights that 2D graphs are typically sparse. This is however not always the case, as there might be many scenarios in which also 2D graphs are dense (for example in social interactions, transaction records, or image regions in images).

- Similar to above, lines 253-256 already hold for many 2D settings, making this argument not solid enough to motivate the need for 3D-specific methods.

- The authors claim that previous methods do not account for the discreteness of the explanation mask. This is however oftentimes addressed by the common design choice of using a Gumbel distribution over the explanation scores (PGExplainer, LRI), which naturally pushes activations close to either 0 or 1. Therefore, **the authors should at least show that their proposed strategy to stimulate discreteness is more effective than using a Gumbel activation**.

- **Only Fidelity- is used to evaluate explanation quality**, whereas other metrics are also available. The authors should at least provide both Fidelity- and Fidelity+, as they are complementary [1,2,3].


[1] Explaining the Explainers in Graph Neural Networks: a Comparative Study. Longa et al. 2024. ACM Comput. Surv.

[2] GraphFramEx: Towards Systematic Evaluation of Explainability Methods for Graph Neural Networks. Amara et al. 2022. LoG

[3] Explainability in Graph Neural Networks: A Taxonomic Survey. Yuan et al. 2023. IEEE

**Questions:**

1. I recommend the authors provide a more solid argument justifying why 3D explanations are radically different from 2D ones (see weaknesses).

2. I suggest verifying that the commonly used Gumbel activation is less effective than the proposed EDMA.

3. Where does the expert knowledge depicted in the last two columns of Figure 4 come from? I suggest adding proper contextualization.

4. Can the authors comment on the differences between GNNExplainer and the proposed solution? To my understanding, the only two differences are: (i) using node scores rather than edges scores; (ii) using two complementary energy scores to promote explanation discreteness, instead of the Gubmel activation or an element-wise entropy as suggested in [4].

[4] Gnnexplainer: Generating explanations for graph neural networks. Ying et al. 2019. NeurIPS

---

### Official Review · Reviewer_R95y · 2024-10-19

**Soundness:** 1
**Presentation:** 2
**Contribution:** 2
**Rating:** 3
**Confidence:** 4

**Summary:**

This paper discusses the challenges and advancements in explaining the decisions of 3D Graph Neural Networks (GNNs), such as in the molecular data. The authors claim that existing explanation methods work well for 2D GNNs but struggle with 3D due to the vast number of edges. The authors propose a new approach to improve explanation accuracy by using energy values to represent an atom's importance in the prediction. By minimizing the error between the predicted and actual subgraphs, their method identifies stable subgraphs with high explanation fidelity.

**Strengths:**

1. This paper uses the energy-based model to provide a constraint rather than the loss function used in GNNExplainer

2. This paper is well-organized, which makes it easy to follow the main idea.

**Weaknesses:**

1. Some core ideas are confusing. For example, the difference between 2D GNNs and 3D GNNs is not clear, even though this paper addresses explanations for 3D GNNs. In Section 3.1, Challenge 1 claims that the random graph is not suitable for 3D GNNs, but it is unclear what kind of assumption should be used. Challenge 2, the dense adjacency matrix, is not a problem in 2D graphs with size regularization.

2. Some expressions are confusing. For example, in Figure 1, row b, the meaning of the red dashed lines and why they are needed is not clear.

3. The metric differs from the mainstream definition, according to [1,2]. The experiments are limited to older methods. It would be better to compare it with more recent methods.


Reference:

[1] Explainability in graph neural networks: A taxonomic survey, TPAMI, 2022

[2] Towards Robust Fidelity for Evaluating Explainability of Graph Neural Networks, ICLR, 2024

**Questions:**

1. What is the difference between the 2D graph explanation results and the 3D graph explanation results?

2. In Figure 4, there are chemical explanations that can be used as ground truth. Can we use the metric to compare with the ground truth?

3. What is the performance of the model to be explained?

---

### Official Review · Reviewer_eCFf · 2024-10-22

**Soundness:** 1
**Presentation:** 2
**Contribution:** 2
**Rating:** 3
**Confidence:** 3

**Summary:**

The authors proposed a novel method for explaining GNN predictions on 3D graphs, such as molecular structures. They introduced a parametrized energy-based function to estimate the probability that a node is important. By adjusting the function’s temperature parameter, they discretize the output probability to either 0 or 1. The method minimizes both the explanation loss and the sum of importance probabilities to identify the most significant subgraph. Experimental results on two prediction tasks from the QM9 dataset, using two GNN models, demonstrate that the proposed method outperforms other baseline approaches.

**Strengths:**

The author identified a key challenge in explaining 3D GNNs: the discrepancy between soft and discrete masks. To address this, the proposed method introduces a novel approach using an energy-based function to represent the importance probability.

**Weaknesses:**

Overall, the proposed method is poorly presented, lacking clear definitions and containing errors in the equations. More detailed explanations and proofs are needed to demonstrate how the two identified problems are effectively addressed. While the use of an energy-based function is new, the method’s novelty is limited, as it primarily introduces a parameterized explainer without significant advancements beyond that.

**Questions:**

1. The second component of Eq. 5 (discrepancy between soft and discrete masks) is not always overlooked, but PGExplainer has already tried to tackle it, which also gives discrete masks (although edge-level masks, but easily changeable to node-level masks).
1. Eq. 7 what is $\bf r$? Why is the specific node i chosen to be included in the loss but not other nodes or the sum of all nodes?
1. Eq. 7, I ignore $\bf r$ and suppose that the second term is the sum of all nodes for now. I don't understand why minimizing the loss in Eq. 7 automatically minimizes the second loss in Eq. 5 (discrepancy between soft and discrete masks). The second term in Eq. 7 means the sum of absolute values of the probabilities of each node being included in the important subgraph. However, minimizing this sum of probabilities doesn't require the probabilities to be discrete, i.e., 0 or 1.
3. It's unclear what is optimized for the loss in Eq. 7 because the optimization problem not explicitly defined. I guess $\zeta, \gamma$? And it's not explained how they are optimized.
2. Eq. 5 is incorrect because left side is graph and right side is loss. You should change to $\min L \le L(...) + L(...)$
4. In row 309 what is $\phi(\cdot)$? It is written that "it extracts logits for class $c$," but where the logit here comes from?
5. Fig. 3 is confusing to me. I guess the line for "pushing energy" means that one choose a small $T$ such that the probability becomes close to either 0 or 1? What does the line of budget mean? I guess it means the user wants to get a subgraph with 2 nodes? Anyways, the figure needs better explanation.
6. Sec 4.4 should be better if the authors can study the method's performance by changing $T$. It seems that the difference between EDMA and EDMA-soft is only the value of $T$. Besides, the values of $T$ for both cases are not given.

---

### Official Review · Reviewer_A7Yw · 2024-10-28

**Soundness:** 3
**Presentation:** 3
**Contribution:** 2
**Rating:** 3
**Confidence:** 4

**Summary:**

The authors propose an improvement to instance-level explanation methods for graph neural networks, specifically for 3D graphs. Their approach involves identifying a subgraph—ideally the smallest possible—that produces a prediction similar to the original graph. While minimizing a loss dependent on a pure mask is challenging (partly due to its combinatorial complexity), previous literature has addressed this by relaxing the mask selecting the subgraph to a soft mask. Once the logits are learned, the subgraph can be obtained by taking max(logits) or selecting values above a certain threshold.

To enhance these methods, the authors introduce an additional loss that pushes the soft masks closer to discrete ones. They achieve this by predicting logits and minimizing a hard concrete-type distribution (eq. 6). This approach encourages the explanation network to predict more "discrete-type" values. The authors also provide an energy-based interpretation of their method.

The authors design experiments to demonstrate that their method outperforms other baselines both quantitatively and qualitatively on the qm9 dataset.

**Strengths:**

The paper is well-written, and the authors effectively introduce and motivate their topic and work. The notation is clear and (mostly) correct, while the figures are well-designed and help clarify the content.

The experiments include both qualitative and quantitative results, as well as an ablation study to assess whether their proposed extra loss term truly affects the method's performance.

The topic is relevant, as explainability remains a significant challenge, especially in domains like chemistry where even domain experts often don't know the ground truth.

The paper offers some novelty, although in my opinion it's rather incremental.

**Weaknesses:**

My main concern is the novelty of the approach (more specific questions about details can be found below in the question section). As I understand it, the authors add a loss term regulating the "discreteness" of the masks. These types of losses are ubiquitous in deep learning; for instance, it's common when learning a permutation matrix to learn a soft-relaxed one and enforce bistochasticity during training.

Furthermore, the interpretation of their method as energy-based, while legitimate, seems rather superficial (every exponential distribution can be viewed as an energy function from a statistical mechanics perspective) and appears aimed at aligning the paper with the current popularity of (generative) energy-based models.

I also find that restricting the evaluation to QM9 is somewhat limited nowadays, as larger datasets like Geom-Drugs and QMugs are becoming standard for assessing new methods. This limitation is particularly notable since the authors claim their method is especially useful for scaling to larger graphs, and QM9 contains only small compounds.

**Questions:**

- Line 269: last equation: indices do not match and \mathbb{m}_i is not defined before (only m_i\in [0,1] is).
- The author mention in several places that the hyperparameters help “managing the budget” K. How is this done in practice? I would expect that K is dynamically chosen by the method, as it is unclear from the beginning how much of a subgraph is required to explain the prediction. But if this is the case, how do we avoid the trivial solution where the whole graph is retained?
- What is the variable \mathbb{r} in equation 7?
- While focused on 3d graphs, the method applies to graph in any dimensions. If the method is indeed superior to others, it should perform at least on par with the SOTA on 2d graphs as well. Further experiments on graph in other dimensions would benefit the paper.
- Figure 4, while very explanatory, it is clearly cherry-picked. It would be helpful also to include further examples, also in the appendix, where the method does not perfectly predict the true chemical explanation.

---

### Official Review · Reviewer_jRWW · 2024-11-02

**Soundness:** 3
**Presentation:** 2
**Contribution:** 2
**Rating:** 3
**Confidence:** 3

**Summary:**

This paper investigates the problem of graph explanation for 3D molecular graphs, specifically predicting which structures within the molecular graph contribute most to the property prediction task. In particular, the graph explanation problem in this paper is equivalent to a mask optimization task. By applying the optimized mask to obscure certain nodes and edges in the graph, the masked graph is then input into the property prediction method. The masked graph that achieves the closest performance to the full graph input represents the most contributive structure. The authors' contribution lies in proposing a new mask optimization method, which uses regularization techniques to make the mask elements closer to 0 or 1, making it easier to identify which parts of the graph should be masked.

**Strengths:**

This model adds a regularization constraint to the soft mask, encouraging the mask values to be closer to 0 or 1, thereby reducing the uncertainty in the output of effective subgraphs. This  sound good.

**Weaknesses:**

The main drawbacks of the paper are three: the motivation is not very clear, the contribution is insufficient, and the experimental section is insufficient. The specific reasons for each are as follows:

1）The motivation is not very clear. First, regarding the motivation of the paper, the authors try to address the Graph Explanation task for 3D molecular graphs. However, both 3D and 2D molecular graphs are essentially graphs; 3D molecular graphs may contain more edges, making the explanation slightly more challenging. Therefore, could the authors provide specific molecular examples or quantitative evidence to demonstrate how existing 2D interpretation methods are inadequate when applied to 3D graphs? Besides, based on the visualization results in Figure 4, the authors are still focused on actual bonds and atoms, which can also be achieved with 2D graph explanation. Interpreting extra edges in a 3D molecular graph defined by a cutoff radius (those not representing actual chemical bonds) is also not particularly meaningful. Therefore, could the authors explain how the proposed 3D molecular explanation method handles extra edges that do not represent actual chemical bonds, and why it is necessary to interpret them?

2）The contribution is not enough. First, using regularization to enhance algorithm performance is a very common approach [1,2]. Furthermore, the authors do not explain the advantages of using the EBM function to design the regularization term. For example, directly applying L1 regularization to the mask in Equation 7 could also make the mask values closer to 0 and 1. Could the authors compare their proposed method with directly applying L1 regularization to the mask and clarify the advantages of using the EBM function for the regularization term? Secondly, the purpose of explanation should be to leverage atom contribution to improve property prediction performance, rather than predicting properties using a smaller subgraph. In the experimental section, the MAE for property prediction using subgraphs is often higher than for the original graph. If you cannot achieve better results than the original graph, then what is the purpose of predicting properties using the optimal subgraph? Thus, the suitability of applying Equations 5 and 1 to molecular property prediction is worth considering. If the goal is simply to explain which parts of the molecule contribute more, the hard mask, as defined in traditional Graph Explanation, may not serve this purpose effectively.

3）Finally, the experiment is insufficient. Molecular research experiments typically involve at least two datasets (such as QM9 and GEOM-Drugs [3]) or more, yet this paper only uses the QM9 dataset, making the experimentation inadequate. Therefore, I suggest that the authors include experiments on the GEOM-Drugs dataset. Additionally, the baselines are outdated, with the latest comparison algorithm from 2022. In the experimental section, the authors need to demonstrate that their 3D interpretation method outperforms 2D interpretation methods from 2024, such as [2,4-9]; otherwise, the proposed 3D interpretation method lacks justification. Therefore, I suggest that the authors include at least one recent graph explanation methods like [2,4,5,7,9] as baselines.

[1] Shan C, Shen Y, Zhang Y, et al. Reinforcement learning enhanced explainer for graph neural networks[J]. Advances in Neural Information Processing Systems, 2021, 34: 22523-22533.
[2] Zhang W, Li X, Nejdl W. Adversarial Mask Explainer for Graph Neural Networks[C]//Proceedings of the ACM on Web Conference 2024. 2024: 861-869.
[3] Axelrod S, Gomez-Bombarelli R. GEOM, energy-annotated molecular conformations for property prediction and molecular generation[J]. Scientific Data, 2022, 9(1): 185.
[4] Huang R, Shirani F, Luo D. Factorized explainer for graph neural networks[C]//Proceedings of the AAAI conference on artificial intelligence. 2024, 38(11): 12626-12634.
[5] Chen T, Qiu D, Wu Y, et al. View-based explanations for graph neural networks[J]. Proceedings of the ACM on Management of Data, 2024, 2(1): 1-27.
[6] Bui N, Nguyen H T, Nguyen V A, et al. Explaining Graph Neural Networks via Structure-aware Interaction Index[J]. arXiv preprint arXiv:2405.14352, 2024.
[7] Liu X, Ma Y, Chen D, et al. Towards Embedding Ambiguity-Sensitive Graph Neural Network Explainability[J]. IEEE Transactions on Fuzzy Systems, 2024.
[8] Homberg S K R, Modlich M L, Menke J, et al. Interpreting Graph Neural Networks with Myerson Values for Cheminformatics Approaches[J]. 2024.
[9] Chen Y, Bian Y, Han B, et al. Interpretable and Generalizable Graph Learning via Subgraph Multilinear Extension[C]//ICLR 2024 Workshop on Machine Learning for Genomics Explorations.

**Questions:**

Additionally, there are some questions regarding the method in the paper:

1) Why was only the QM9 dataset used in the experimental section, and not the GEOM-Drugs dataset? Typically, studies on molecules utilize both of these datasets.

2) How does the soft mask extract the optimal subgraph? Is it determined by setting a threshold manually?

3) There are some issues with Equation 7. What does ‘r’ represent in the equation? Also, for the energy function term, isn't it necessary to sum over all nodes? Additionally, Equation 7 does not specify the parameters to be optimized. Is the EBM function the only part that requires optimization?

4) I suggest adding a flow figure to illustrate the method and training process.

5) In Section 4.3, you presented visualization results based on DimeNet++ backbone. If  different backbone, such as SchNet, is used, would the visualization results be the same? Does Graph Explanation only depend on the molecule itself, or is it influenced by the network architecture as well?

6) Why are both a multigraph (with two edges appearing between the black and white atoms in the leftmost image) and an adjacency matrix shown in Figure 4(b)(c)? How can the adjacency matrix represent the adjacency relationships in a multigraph?

---

### Meta-Review · Area_Chair_uYnw · 2024-12-22

**Metareview:**

The paper adopts graph explanation techniques to attribute molecular properties to substructures within the 3D molecular graph. It approaches the problem as a mask optimization on the molecular graph for the interested property prediction. It regularizes a continuous mask to take values closer to binary values for better interpretability.

The reviewers acknowledged the importance of the problem and the relevance of the introduced regularization but were concerned about the lack of enough technical novelty or significant empirical results/insights, the insufficient experimental support of the method on different benchmarks, the absence of comparison to several graph explanation techniques, and the unclarity of writing and presentation at several occasions.

The reviewers unanimously recommend rejection and the authors did not provide a rebuttal. The AC does not find any reason to overcome the unanimous suggestion and thus recommends rejection.

**Additional Comments On Reviewer Discussion:**

The paper was reviewed by a panel of five experts in graph networks, explanations, and bioinformatics. They all leaned towards rejection initially. Authors did not provide a rebuttal and the reviewers kept their rating.

---

### Decision · Program_Chairs · 2025-01-22

Reject